

# Assessing the risk posed by natural hazards to infrastructures

Unni Marie Kolderup Eidsvig[1], Krister Kristensen[1], and Bjørn Vidar Vangelsten[1]
(1) NGI, Oslo, Norway

* Correspondence to: Unni M. K. Eidsvig at Unni.Eidsvig@ngi.no.

**Abstract.** This paper proposes a model for assessing the risk posed by natural hazards to infrastructures. The model prescribes a three level analysis with increasing level of detail, moving from qualitative to quantitative analysis. The focus is on a methodology for semi-quantitative analysis to be performed at the second level. The purpose of this type of analysis is to perform a screening of the scenarios of natural hazards threatening the infrastructures, identifying the most critical scenarios and investigating the need for further analyses (third level). The proposed semi-quantitative methodology considers the frequency of the natural hazard, different aspects of vulnerability including the physical vulnerability of the infrastructure itself and the societal dependency on the infrastructure. An indicator-based approach is applied, ranking the indicators on a relative scale according to pre-defined ranking criteria. The proposed indicators, which characterize conditions that influence the probability of an infrastructure break-down caused by a natural event, are defined as 1) Robustness and buffer capacity, 2) Level of protection, 3) Quality/Level of maintenance and renewal, 4) Adaptability and quality in operational procedures and 5) Transparency/complexity/degree of coupling. Further indicators describe the societal consequences of the infrastructure failure, such as Redundancy and/or substitution, Restoration effort/duration, Preparedness, early warning and emergency response and Dependencies and cascading effects. The aggregated risk estimate is a combination of the semi-quantitative vulnerability indicators, as well as quantitative estimates of the frequency of the natural hazard, the potential duration of the infrastructure malfunctioning (depending e.g. on the required restoration effort) and the number of users of the infrastructure.

Case studies for two Norwegian municipalities are presented where risk posed by adverse weather and natural hazards to primary road, water supply and power network is assessed. The application examples show that the proposed model provides a useful tool for screening of potential undesirable events, contributing to a targeted reduction of the risk.

*Keywords: Infrastructure risk, infrastructure vulnerability, natural hazards, extreme weather events*

## 1 Introduction

The modern society is increasingly dependent on infrastructures to maintain its function and disruptions in one of the infrastructure systems may have severe consequences. With a changing climate, the frequency and intensity of some extreme





weather events (e.g. intense precipitation) and related hazards are expected to increase, (Montesarchio and Herrere, 2015; Hanssen-Bauer et al., 2015), creating challenges for the infrastructures.

Since the financial and workforce resources available to operators to protect their infrastructure systems are limited, it is especially important to use resources efficiently and effectively. To do so, it is essential to be aware of the threats and risks and of the possibility to compare and assess risk in order to set priorities. This will be the basis for implementing targeted protection measures, as stated by Federal Ministry of the Interior (2008).

The main purpose of performing risk assessment of infrastructure affected by adverse weather and natural hazards is to support decision making, to provide information to policy development and to support well-founded risk management. An effective risk assessment is indispensable in order to identify threats, vulnerabilities and evaluate the impact on critical infrastructures, taking into account the probability of the occurrence of these threats. The risk assessment gives decision-makers a better understanding of risks and its uncertainties, describing and comparing the sensitivity, vulnerability and potential risks related to the effects on critical infrastructures from natural events.

Careful assessment of potential risk and informed analysis of dependencies between infrastructures can significantly contribute to effective investment in planning and design and facilitate preparedness actions in the event of failure. Optimal decisions require that decision makers are aware of how their decisions may affect the expected loss. Understanding the severity of risk can also be used to determine if an infrastructure (or parts thereof) is deemed to be critical, (McCord et al., 2015).

In general, the purpose of risk assessment related to infrastructures is to find answers to the following questions (Räikkönen and Tagg, 2015):

1. What are the threats and hazards of relevance that the infrastructures under consideration might be exposed to? (Danger identification)

2. How often does, or how likely is it for an adverse event to occur? (estimation of likelihood of occurrence; hazard)

3. What can go wrong? (Evaluation of sensitivity (susceptibility) and resilience, resulting cascades and consequences)

4. How bad are the consequences? (Assessment of severity of consequences and risk, identification of high consequence scenarios)

After the risk assessment is carried out, the final questions are: "What should be done? What are the obvious and non-obvious vulnerabilities and how can they be reduced and/or better managed?". This question requires a discussion about acceptability/tolerability of risk and the potential mitigation measures, cost/benefit assessment, prioritisation and urgency.

In order to target the scope of the risk assessment, it is favourable to perform analyses at different levels, starting with a coarse analysis and subsequently increase the degree of detail. The coarsest analyses are usually done by subjective assessments and considerable use of expert judgment. For such analyses broad expertise is essential to ensure satisfactory quality of the analysis




results, otherwise the assessment may be too coarse, important events may be overlooked and the results of the assessment too subjective. On the other hand, detailed quantitative analyses are often too complex and time-consuming to be applied as a tool to identify the most critical risk scenarios. An alternative tool for screening of the potential scenarios in a systematic, transparent and repeatable way could bridge this gap. This paper propose an explicit methodology applicable for such

screening.

## 2 Scope

The aim of the work in this paper is to propose a comprehensive and user-friendly method for identification and assessment of natural events/natural hazards leading to malfunctioning of infrastructure. The method is designed to be consistent with, and a supplement to, the guidelines for municipal risk and vulnerability analysis in Norway, provided by the Norwegian directorate

for Civil Protection, DSB (2014). The proposed method aims to be applicable within the main infrastructures (electricity supply, water supply, transportation, and ICT) and to provide support for mapping of threats from natural events, for planning and preparedness and for prioritization of risk reduction measures. Strategies for risk reduction fall into two types: those that minimise the probability of infrastructure failure, and those that minimise the negative effects of a failure, (IRGC, 2007). The proposed method takes into account the vulnerabilities of infrastructure and barriers that affect the probability of infrastructure

failure. It also considers factors affecting the social consequences of malfunctioning of the infrastructure. The scope is schematically illustrated in Figure 1.

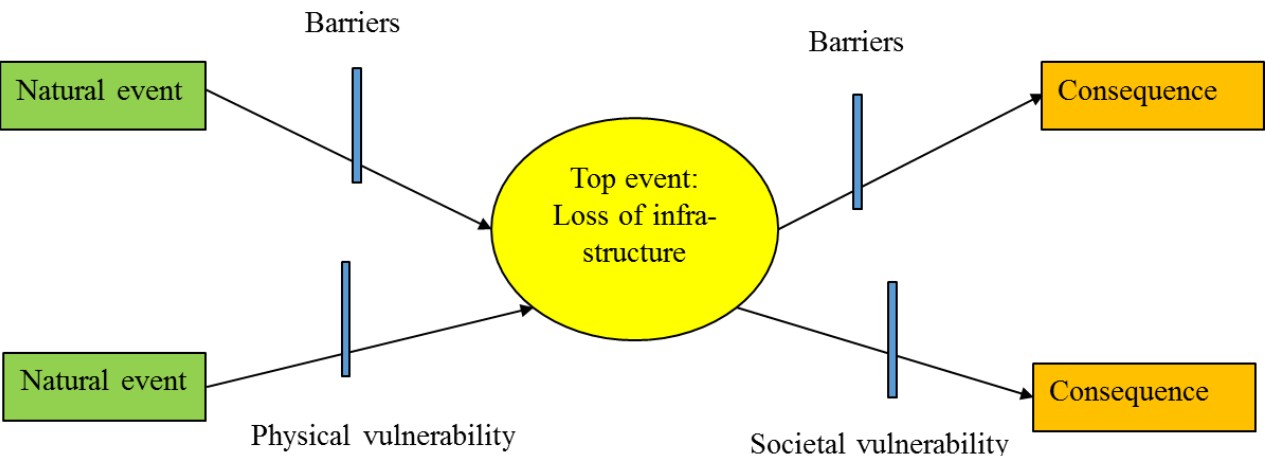

**Figure 1. Schematic representation of the scope. Factors that affect the probability of the top event are shown on the left side (i.e.**
**causes, barriers and physical vulnerability of the infrastructure). Factors that affect the concequences of the top event are shown on the right side.**



## 3 Background

By law, the Norwegian municipalities are required to carry out a risk and vulnerability analysis and plan and prepare for emergencies in a short- and long-term perspective. The purpose with the duty/legislation is to ensure that the municipalities are working holistically and systematically with societal safety and preparedness across sectors in the municipality. Knowledge about risk and vulnerability is important to reduce the probability of undesirable events and to reduce the consequences should the event occur. The current format of the municipal risk and vulnerability assessments is very similar to a preliminary hazard analysis (PHA) where the starting point is the identification of undesired events, followed by a simple probability and consequence assessment of each event. Through the work with the risk and vulnerability analysis, the municipality obtains a better overview over, and an increased consciousness about, the relevant risks and vulnerabilities. In addition, the municipality can acquire knowledge about how risks and vulnerabilities can be managed. The ultimate goal of the analyses is to ensure the safety of the inhabitants. This goal is further specified through four societal values with corresponding consequence types as shown in Table 1.

Vulnerability analysis of the infrastructures and their interdependencies is an essential part of the municipal risk and vulnerability analysis for the societal value named "Stability", i.e. the consequences like "Lack of basic provisions" and "Disruptions in daily life".

A literature review on vulnerability assessment of critical infrastructure and on losses caused by adverse weather and natural hazards has been conducted in this study. Infrastructures have some basic traits in common, such as large-size, wide-area coverage, complexity and interconnectedness, but show significant differences in detail. Methods for vulnerability assessment vary with the type of system, the objective of the analysis, the analysis steps and the available information. No all-encompassing method exists, but rather an interplay of methods is necessary to provide trustworthy information about vulnerabilities within and among critical infrastructures (CIs), including the effect of (inter)dependencies (Kröger and Zio, 2011). Methods used for vulnerability and risk assessment of infrastructure include susceptibility functions, economic theory based approaches, probabilistic modelling, statistical analyses of past events, empirical approaches, risk analysis of technological systems, network based approaches, agent based approaches, system dynamics based approaches, relational databases and use of vulnerability and risk indices. Meyer et al. (2013) give a broad review of assessment of costs of natural hazards (considering both direct and indirect costs). Yusta et al. (2011), Kröger and Zio (2011) and Ouyang (2014) provide extensive reviews of vulnerability and risk assessment of infrastructure. Solano (2010) reviews and evaluates methodologies to assess vulnerabilities of critical infrastructures across a number of characteristics. Rinaldi et al. (2001) provide an overview of how to identify, understand and analyse interdependencies between infrastructures.

In the following, special attention is given to methods that apply vulnerability and risk indices or identify factors relevant for vulnerability and risk for infrastructures affected by adverse weather and natural hazards, in particular those of Federal Ministry



of the Interior (2008), Lenz (2009), Merz (2008), Vatn (2009) and Kröger (2008). The German Federal Ministry of the Interior (2008) provides guidelines for operators of critical infrastructures, providing a management strategy to identify risks, implement preventive measures and handle crises effectively. Lenz (2009) provides a detailed overview of the vulnerability of critical infrastructures, distinguishing between indicators relevant for vulnerability of critical infrastructure and for coping

capacity. Merz et al. (2010) go through various aspects of the assessments of economic flood damage. Vatn et al. (2009) has developed a methodology that identifies adverse events as well as risk and vulnerability factors which may affect the likelihood and consequences of undesirable events. Kröger (2008) discusses the most significant factors related to the risks faced by critical infrastructures. These include societal, system-related, technological, institutional and natural factors, with a special focus on issues associated with the increasing interdependence between infrastructures. The indicators in the abovementioned

literature, identified as the most important for the scope of this paper, are summarized below:

**Dependencies:** Dependencies of other infrastructures, specific personnel and specific environmental conditions to work makes the infrastructure more vulnerable; Federal Ministry of the Interior (2008), Vatn et al. (2009), Lenz (2009) and Kröger (2008).

**Robustness:** The physical robustness of risk elements (in particular facilities, equipment, buildings) is an important factor for whether they will be damaged by an extreme incident; Federal Ministry of the Interior (2008), Lenz (2009)

**Buffer capacity:** Buffer capacity means that the sub-process can tolerate the effects of an incident to a certain degree and for a certain time without being affected; Federal Ministry of the Interior (2008), Lenz (2009)

**Level of protection:** A risk element not sufficiently protected against a threat is vulnerable should this threat arise; Federal Ministry of the Interior (2008), Lenz (2009)

**Quality level/Level of maintenance and renewal**: To ensure appropriate quality of the infrastructure it need to be maintained

and renewed systematically; Lenz (2009), Vatn (2009)

**Adaptability:** Ability to adapt to changing framework conditions makes the infrastructure less vulnerable; Federal Ministry of the Interior (2008).

**Quality in operational procedures**: The vulnerability of the infrastructure depends on how well it is operated; Vatn (2009); Kröger (2008).

**Transparency/complexity/degree of coupling:** The complexity of the infrastructure and its dependency on single components to work contributes to higher vulnerability; Perrow (1984), Federal Ministry of the Interior (2008), Vatn (2009), Kröger (2008).

**Redundancy/substitutes:** If there is an outage or reduced capacity in the infrastructure, it is easier to handle if there are back-ups or substitutes to the infrastructure; Federal Ministry of the Interior (2008), Vatn (2009), Lenz (2009).

**Restoration effort/duration:** Restoration effort refers to the effort needed to restore a damage risk element including monetary costs as well as time and staff resources needed; Federal Ministry of the Interior (2008), Vatn (2009), Lenz (2009).

**Preparedness:** An outage of an infrastructure is easier and more quickly restored or better handled if the situation has been prepared for; Lenz (2009), Vatn (2009), Merz (2010)





**Early warning, emergency response and measures:** If the warning time is sufficiently long, an early warning system combined with emergency response and measures may reduce the consequences of an infrastructure outage; Merz (2010).

**Cascading effects and dependencies:** The definition and content of the term cascading effects are discussed by Pescaroli and Alexander (2015) and in short referred to as "chain-sequence of interconnected failures" or as second-order/higher order effects; (Rinaldi et al.; 2001). Cascading effects and dependencies of other societal functions on the infrastructure increase the consequences of the infrastructure loss; Vatn (2009), Federal Ministry of the Interior (2008), Lenz (2009).

## 4 Methodology

The method prescribes a three level analysis with an increasing degree of detailing and quantification:

- Level 1: Qualitative: risk identification
- Level 2: Semi-quantitative analysis to rank the risk: Screening of the scenarios of natural hazards threatening the infrastructures (identified in the level 1 analysis) in which the scenarios with potential highest risk are identified
- Level 3: Quantitative analysis: detailed analysis of the scenarios identified in the level 2 analysis.

The presented work focuses on the second level, i.e. a semi-quantitative analysis or a ranking of the risk. A mixture of a quantitative approach and an indicator-based approach is chosen for the purpose. The choice of indicators is based on findings from the literature as presented in Section 2. As illustrated in Figure 1, the risk governed by casual factors, influencing the probability of the malfunctioning of the infrastructure as well as factors relevant for the societal consequences of the malfunctioning infrastructure. The indicators are grouped into two main groups accordingly. Vulnerabilities of infrastructure that affect the probability of failure in the infrastructure are referred to in the following as physical vulnerabilities. The likelihood of infrastructure failure is also affected by barriers. Vulnerabilities that affect the social consequences are referred to as societal vulnerabilities. The indicators chosen for assessment of the physical vulnerability (including barriers reducing the probability of the malfunctioning of the infrastructure) applied in this method are:

- Robustness and buffer capacity
- Level of protection
- Quality level/Age/Level of maintenance and renewal
- Adaptability and quality in operational procedures
- Transparency/complexity/degree of coupling

The number of indicators were reduced compared to the indicators listed in the background section: Robustness and buffer capacity were combined since they are closely related; but with the difference being that buffer capacity also deals with the temporal aspect. Furthermore, adaptability and quality of operational procedures were merged into one indicator. Adaptability is related both to the adaptations that are physically possible, but also to the quality and timing of the practical implementation of adaptation. Adaptability therefore also depends on how the infrastructure is operated. Grothmann et al. (2013) discuss and





compare frameworks for adaptive capacity for institutions. The indicators for dependencies on external factors for the infrastructure to work would typically also be among the physical vulnerability indicators. These are however omitted here, as they are considered less relevant for loss of infrastructure caused directly be natural events, and thus outside the scope of this method. (The method does not consider infrastructure outages caused by loss of other infrastructures or by lack of resources.)

5   The chosen indicators for the societal vulnerability in this study thus include the following:

- Redundancy/substitutes
- Cascading effects and dependencies
- Preparedness
- Early warning, emergency response and measures

The duration of the infrastructure malfunction is included quantitatively in the consequence assessment, see Figure 2 and Table 3. Thus, the indicator "Restoration effort/duration" is omitted here to prevent it from being taken into account twice in the risk assessment.  Figure 2 illustrates the decomposition of the risk and the grouping of the indicators based on their relevance for the frequency of the infrastructure malfunctioning (caused by natural events) and the corresponding societal consequences.

15   The Figure also illustrates the content of the explicit proposed method for semi-quantitative risk assessment. The next subsection presents a more detailed description of the analysis steps of that method.





**Figure 2. Illustration of the method for semi-quantitative analyses. The indicators with bold frames are assessed quantitatively for an initial categorization of the probability and consequence.**

5   **4.1 Methodology for semi-quantitative risk assessment (level 2 analysis)**

The method propose to perform the semi-quantitative risk assessment in three steps:

Step 1: Initial categorisation of the probability and consequence of the top event (natural hazards causing malfunctioning of the infrastructure).

Step 2: Vulnerability assessment: Ranking of the vulnerability indicators, estimation of the physical and societal vulnerability

10   scores.

Step 3: Final categorization of probability and consequence, based on the initial categorization and results from the vulnerability assessment



The content of each of the steps will be outlined in the following.

**Step 1: Initial categorization of probability and consequence of the top event (natural hazards causing malfunctioning of the infrastructure)**

In the initial probability classification, the analyst need to assign the probability of the natural event into one of five quantitatively defined probability categories. The categories range from an annual probability lower than 0,1% (probability category A) to an annual probability higher than 10% (probability category E). Table 2 shows the scheme for the categorization into categories A – E. These probability categories correspond to the categories suggested in DSB (2014).

In the initial consequence categorization, the analyst needs to assign the consequences to one of five consequence classes. In this step, the consequences are determined by the combination of duration of the infrastructure malfunctioning and the numbers of users served by the infrastructure. The lowest consequence category (consequence category 1) corresponds to relatively few users combined with short duration, while the highest consequence category (consequence category 5) corresponds to relatively many users combined with a long duration of malfunction. Table 3 shows the scheme for the categorization of consequence into consequence categories 1-5.

**Step 2: Vulnerability assessment: Ranking of the vulnerability indicators, estimation of the physical and societal vulnerability scores**

The vulnerability assessment is performed using an indicator-based approach. This type of approach enables the combination of information from different sources and different formats, e.g. qualitative and quantitative data. The chosen vulnerability indicators for the method are listed previously in this section and illustrated in Figure 2. The indicators are grouped into physical vulnerability indicators and societal vulnerability indicators. Firstly each of the vulnerability indicators are assigned a score value on the scale 1 – 5, where 1 implies low vulnerability and 5 implies high vulnerability. Next it is beneficial, both for the sake of simplicity and in order to formulate user-friendly explicit procedures, to estimate one aggregated physical vulnerability score and one aggregated societal vulnerability score. There are different ways of performing such a combination. Approaches for combining the indicators may be to e.g. estimate arithmetical or geometric averages, to perform a fuzzy set analysis or to apply a multi criteria decision approach. In this paper it is chosen to aggregate the indicator scores into a physical vulnerability score estimated as a weighted average of the individual score of the physical vulnerability indicators and a societal vulnerability score estimated as a weighted average of the individual score of the societal vulnerability indicators. The weights will vary with the scale, type and importance of the infrastructure in study and are to be chosen by the user of the method. The flexibility to adjust the weights, combined with the generic formulation of the indicators, make the method suitable to different types of infrastructures and different types of natural events. All the steps of the procedure are implemented into an Excel-format work sheet to provide a simple and user-friendly tool for the risk assessment.

 a) Physical vulnerability assessment: To begin with, score values 1-5 need to be assigned to each of the physical vulnerability indicators. A choice of score value 1 implies that the analysed infrastructure has an optimal realization with respect to the analysed indicator, which means that the physical vulnerability of the infrastructure is low with



respect to the indicator. Optimal realizations of indicators imply properties of the infrastructure such as a high robustness and high buffer capacity, high level of protection against the analysed natural event, high quality level, new or very well maintained infrastructure, a high degree of adaptability and quality in operational procedures, a high degree of transparency and that the infrastructure system is has a manageable degree of complexity and coupling.

Score value 5 implies the analysed infrastructure has a severe weakness with respect to the analysed indicator, which means that the indicator contributes to a high physical vulnerability. The criteria chosen to describe the physical vulnerability for each indicator are outlined in Table 4. After the scoring of the indicators, the physical vulnerability score is estimated as described above.

b) Societal vulnerability assessment: Initially, the score values 1-5 need to be assigned to each of the societal vulnerability indicators. A choice of score value 1 implies that the society has an optimized solution with respect to the analysed indicator and infrastructure, contributing to lower societal vulnerability. This is the case if the society have parallel systems to the infrastructure or substitutes that could offer the same services as the analysed infrastructure, if the infrastructure is less important for the society and the malfunctioning is not associated with

potential cascading effects and that there are routines for preparedness and emergency response to mitigate the consequences. Score value 5 implies that the society is especially vulnerable to malfunctioning of the infrastructure, with respect to the analysed indicator, i.e. the indicator contributes to a higher societal vulnerability. The criteria chosen to describe the societal vulnerability for each indicator are outlined in the scheme in Table 5. After the scoring of the indicators, the societal vulnerability score is estimated as described above.

**Step 3: Final categorization of probability and consequence, based on the initial categorization and results from the vulnerability assessment:**

The physical vulnerability score is used to adjust the probability category assessed in the initial categorization and the societal vulnerability score is used to adjust the consequence category assessed in the initial categorization, as described below.

a) Adjustment of probability category: The physical vulnerability score is seen as a proxy for the probability of a break-down in the infrastructure, assuming that the natural event in the study has occurred. Thus, low physical vulnerability means that the probability that the infrastructure will break down (due to the natural event) is assumed to be much lower than the frequency for the natural event (for instance one order of magnitude). On the other opposite, if the

30 physical vulnerability is very high, the probability that the natural event will cause infrastructure malfunctioning is also very high. The the probability that the infrastructure will break down (due to the natural event) is similar to the natural event. This conditional probability is accounted for by adjusting the probability category chosen in the initial categorization. The physical vulnerability score is applied to adjust the probability category according to suggested criteria shown in Table 6. However, judgment should be used when applying these criteria, taking into account, e.g.




whether the probability of the natural event belongs to the lower range within the category or to a higher range. In addition, whether one of the vulnerability indicatorsis considered as having higher importance than the others in the analysed case.

b) Adjustment of the consequence category: The societal vulnerability score is a measure of the ability of the society to
5     keep up its functions without the specific infrastructure. If the societal vulnerability is very low, the society is able to keep up its functions despite the malfunctioning infrastructure, thus the initial consequence category could be an overestimation of the consequences. Accordingly, the final consequence category should be adjusted down from the initial consequence category. If the societal vulnerability score is high, the number of affected people can exceed the number of infrastructure users and the final consequence category could be higher than the initial one. The societal
10     vulnerability score is applied to adjust the consequence category according to the suggested criteria in Table 7.

When all the steps are performed, each analysed scenario is assigned a probability category (A-E) and a consequence category (1-5). The risk level is determined by the combination of these, subdivided into 7 risk levels as shown in Table 10.

## 5 Application examples for the municipalities Stryn and Hornindal

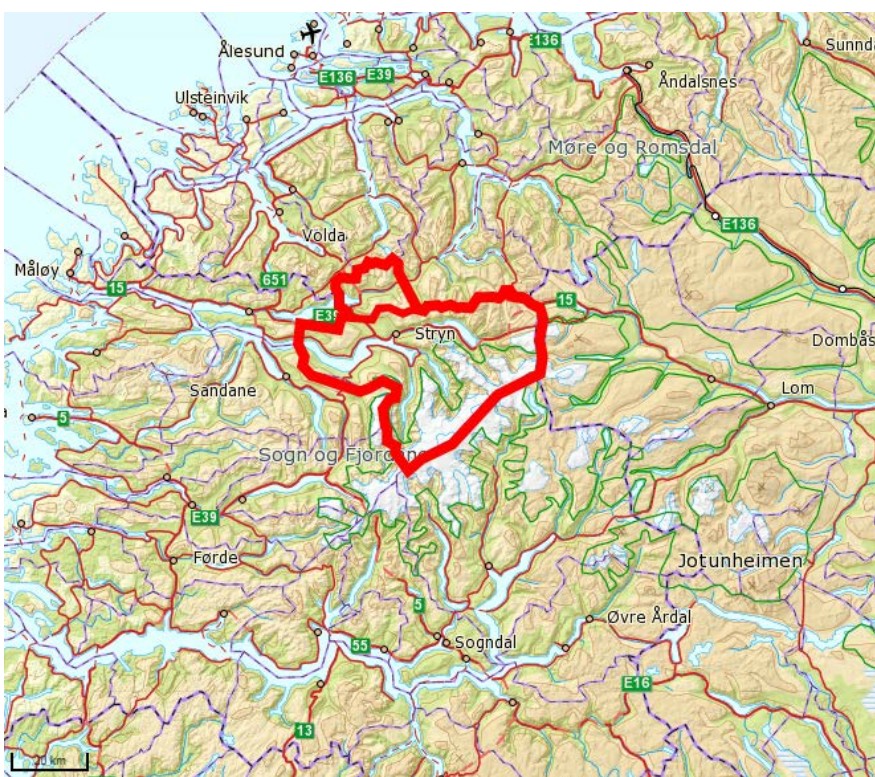

**Figure 3 The Stryn and Hornindal municipalities, Western Norway**





Stryn and Hornindal are municipalities in the county Sogn og Fjordane in West Norway. The characteristics for the area are the combination of fjords, glaciers, rivers and water. There are high and steep mountains, deep valleys and rich valley floors and valley sides with good growing conditions. The municipalities are situated at coastal side of the water divide between coastal and inland climatic zones, with strong orographic effects on precipitation and weather. The industries are varied, but consist mainly of small and medium size industrial establishments. On the main roads there is a rather high proportion of utility transportation.

The following generic scenarios for Stryn and Hornindal were identified as potential undesirable events, based on their qualitative municipal risk and vulnerability analysis, Stryn og Hornindal kommuner (2014):

- Landslides threatening power distribution, water supply or transportation
- Floods contaminating drinking water
- Ice jam breakup in river leading to failure in sewage system
- Storm leading to closed roads or ferry services
- Storm leading to failure in power supply and communication

These scenarios were further concretized into the following site-specific scenarios:

1. Snow avalanche against main road RV 15 at Strynefjellet
2. Debris flow against Innvik waterworks
3. Snow avalanche across main road 724 to Oldedalen
4. Storm leading to failure in electricity distribution and communication to the municipal centre
5. Landslide across main road E39 at Skredestranda
6. Ice jam breakup in Storelva-river in Hornindal, failure in sewage system
7. Storm leading to closure of the ferry service between Anda and Lote

The location of these scenarios are indicated in Figure 4.




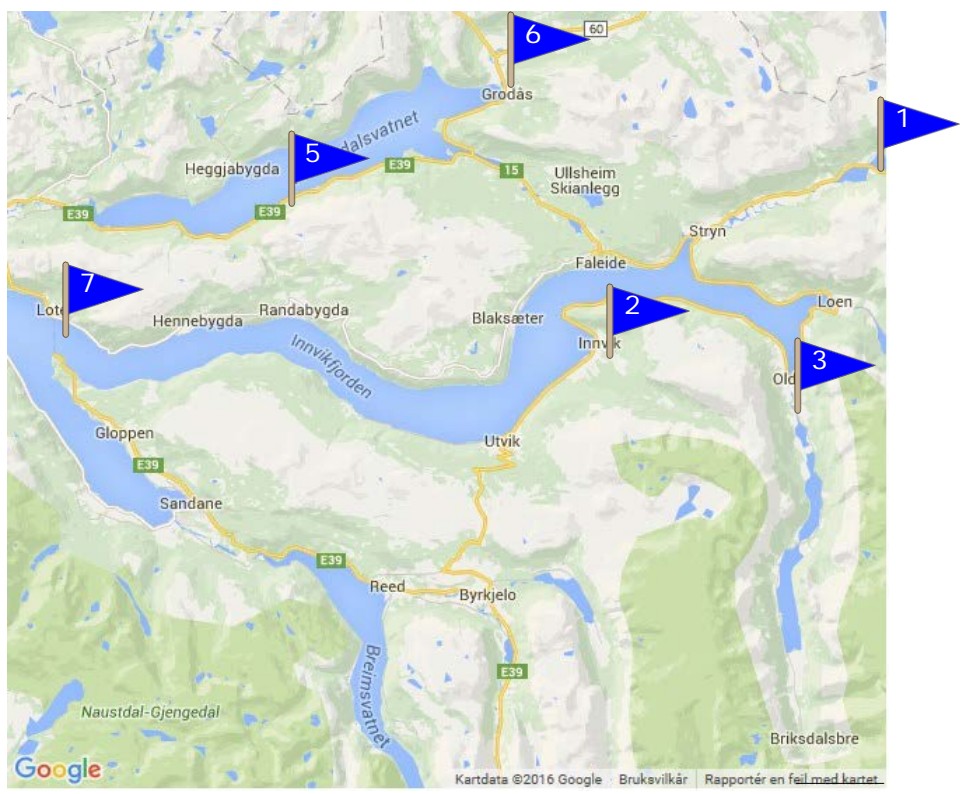

**Figure 4 Overview of the locations of the site specific scenarios. The location of each scenario is identified with a blue flag with a number, referring to the scenario number in the list above. (Flag 1 shows the location of RV 15 to Stryn, but the actual scenario is located at a part of the road outside the map)**

**5.1 Results**

The method was applied for the 7 scenarios listed above. The ranking of the vulnerability indicators for each of the scenarios are presented in Table 8. The initial and final categorization of probability and consequence, as well as the basis for the categorization (i.e. the frequency of the natural event, the duration and number of people served by the infrastructure) are shown in Table 9. Explanation to and reasoning for the ranking is given in Appendix A. The method has been implemented in an Excel sheet in which the ranking, weighting and calculations have been performed.

The results of the analyses are placed in a matrix with increasing severity of consequence along the first axis and increasing probability along the second axis. The corresponding risk level is determined by location in the matrix, subdividing the risk into 7 risk levels illustrated with colour codes. In this way the risk associated with each of the scenarios could easily be compared and the most critical scenarios identified. The risk matrix with the results of the analyses for the 7 scenarios are shown in Table 10.



As Table 10 shows, the ranking of the risk associated with the analysed scenarios is as follows:

- Risk level 7: Storm leading to failure in electricity distribution and communication to the municipal centre, Landslide across main road E39 at Skredestranda
- Risk level 6: Snow avalanche against main road RV 15 at Strynefjellet, Snow avalanche across main road 724 to Olderdalen
- Risk level 5: Debris flow against Innvik waterworks
- Risk level 4: Ice jam breakup in Storelva-river in Hornindal, failure in sewage system, Storm leading to closure of the ferry service between Anda and Lote

None of the analysed scenarios ended up being low risk scenarios. This is not a coincident, as the selected scenarios are based on generic scenarios identified in Stryn og Hornindal kommuner (2014) where considered plausible scenarios posing risk to the municipalities were selected. In addition, in order to facilitate the data collection for the site specific scenarios, scenarios that had already occurred were applied to demonstrate the application of the model.

The results of the analyses provide a better overview over the relevant risks and vulnerabilities and contribute to an increased consciousness in the municipalities. Knowledge about risk and vulnerability associated with the identified scenarios is an important first step to reduce the risk, risk reduction is especially important for the scenarios with the highest risk, e.g. at risk level 6 and risk level 7. All the three scenarios with landslide or avalanche across roads emerge as the most critical scenarios, in addition to the failure in electricity and communication caused by storms. The risk could either be reduced by reducing the probability of the scenario (e.g. through implementation of physical mitigation measures for landslides on the most exposed parts of the road) or to reduce the associated consequences (e.g. through an improvement of the societal vulnerability indicators, such as establishing redundant infrastructure systems). By systematic work with municipal risk analyses followed by associated risk management actions and repeated over time, the municipality can move step by step towards increased safety and stability for the inhabitants.

## 6 Discussion

The focus of the method described in Section 4 is to propose a tool for screening of the potential scenarios in an explicit, systematic, transparent and repeatable way that could be applied at the intermediate level in the three level approach. It is a complicated and labour intensive task to analyse the risk associated with infrastructure systems and we see the three level strategy as a practical approach to target the scope of the risk assessment and minimise the analysis effort. The proposed method serves as an alternative to other risk assessment methods with low to intermediate precision and resolution. It provides more guidance to the user than general risk assessment methods and could therefore be used also by non-experts on risk assessment, for example by stakeholders in Norwegian municipalities. The method guides the user in a systematic way through the risk assessment of infrastructure affected by natural events with related consequences for the society. The guidance is provided through vulnerability indicators in terms of the vulnerability of the infrastructure itself and its importance in society.





Application of the method assigns a relative risk level to each of the scenarios, where risk level 1 implies the lowest risk and risk level 7 the highest risk. The method produces results that are easy to communicate and provides a good communication tool. The indicator-based approach for the vulnerability assessment enables a combination of different types of data from different sources and knowledge domains and on different formats. The method does not require a large amount of data, but

5 does require that the user has comprehensive knowledge about the local conditions, properties of the infrastructure, how the infrastructure is operated, is aware of the hazard situation in the area with respect to natural events and is capable of assessing the frequency of the hazard and the importance of the various vulnerability factors for the infrastructure being studied. The initial categorization of probability and consequence governs the outcome of the risk analysis, which is dependent on the background knowledge of the user.

The proposed method provides, in addition to the risk ranking, implicit guidance on how to reduce the vulnerability (through the information in each of the indicators), and consequently also the risk. The methodology is comprehensive, yet fast, and is designed to be consistent with, and a supplement to, the guidelines for municipal risk and vulnerability analysis in Norway, provided by the Norwegian directorate for Civil Protection, DSB (2014). The proposed method aims to be applicable within

15 the main infrastructures (electricity supply, water supply, transportation, and ICT) and to provide support for mapping of threats from natural events, for planning and preparedness and for prioritization of risk reduction measures.

It should be noted, however, that the accuracy of the method is lower than for a purely quantitative assessment and could not be immediately used for cost-benefit analyses of mitigation strategies. Ideally, the method should be calibrated against quantitative data. This could partly be done for the probability of infrastructure malfunctioning by using historical data from

20 infrastructure operators on infrastructure malfunctioning combined with the properties of the infrastructure. For the societal factors, however, a calibration is more difficult as the societal ability to cope without the analysed infrastructure is not an observable quantity (or could be linked to other observable quantities) that could be relevant to compare with.

## 7 Conclusions

This paper shows the development and demonstration of a method for screening of scenarios posing potential high risk in terms of stability for the local society in accordance with the Norwegian guidelines. The method is intended to be the second level of a three-stage methodology for risk assessments, where level 1 consists of risk identification and level 3 consists of detailed quantitative analysis. While the proposed methodology could be applied for all types of natural events and all types of infrastructures, level 3 analyses will to a larger extent need to be adapted to the types of the specific infrastructures and

hazards. The analysis may be part of a municipal risk and vulnerability analysis. It can be used on different scales by adapting the consequence categories and can be adapted to different infrastructures through the flexible weighting system.



The indicator approach and less ambiguous ranking criteria for the physical as well as the societal indicators make the model easy to use for people knowledgeable of the municipality and its infrastructures. The proposed method is seen as a useful screening tool for identification of the most critical scenarios and produce results that are easy to understand and to communicate.

Assessment of potential threats and their related risks, including identification of the most critical scenarios, is essential to set priorities for infrastructure protection. The risk assessment contribute to targeted investment in planning and design and facilitate preparedness actions in the event of failure.

**Appendix A**

The description of the assessment of each scenario as well as explanation of the ranking is given in the following subchapters.
Some of the identified scenarios are scenarios that have already occurred and are expected to occur again. Other scenarios have not occurred, but were considered plausible. For the already occurred scenarios, observations and newspaper reports were used as data sources to support the ranking of the indicators.

**Scenario 1: Snow avalanche against main road RV 15 at Strynefjellet**

The ranking of this scenario is based on Kristensen (2005), observations from the area, records of previous events and expert
judgment. Selected ranking score for each of the scenarios are given in parentheses.

Probability:

- *Frequency of natural hazard:* Every 5 years, i.e. 0.2 per year for the largest snow avalanche. This corresponds to the probability category "E" in Table 2.

Vulnerability assessment:

- *Robustness and buffer capacity:* The road will be closed in case of high avalanche danger. (4)
- *Level of protection:* Some parts of the road are especially exposed to snow avalanches because of the lack of any physical protection (5)
- *Quality level/Age/Level of maintenance and renewal:* The road is relatively old, but is satisfactorily maintained (3)
- *Adaptability and quality in operational procedures:* The infrastructure is operated by an experienced operator (2)
- *Transparency/complexity/degree of coupling:* Relatively low degree of complexity and coupling. (2)

Societal consequences:

- *Number of infrastructure users:* The annual daily traffic (ADT): 800
- *Duration:* Good routines for clearing of the road. Large avalanche: duration 2 days, small avalanches: duration 8 hours. The duration can be longer if the road is closed because of avalanche danger or in combination with adverse
weather.

The abovementioned combination of users and duration qualify for consequence category 3-4 according to Table 3.



- *Redundancy/substitutes:* Alternative roads offer long diversions on partly avalanche exposed roads. (4)
- *Cascading effects and dependencies*: Moderate cascading effects, mainly economic consequences as there is a high proportion of utility transportation on the road. (3)
- *Preparedness:* Very high risk awareness and high level of preparedness. (1)
- *Early warning, emergency response and measures:* Early warning and closure of the road can act as a measure to save human lives, but does not prevent the economic consequences of the road closure. (5)

**Scenario 2: Debris flow against Innvik waterworks**

The ranking is based on a similar historic event in 2014; information given on the homepage of the municipality http://innvik.vikanenett.no/, in the reports from DSB (2015) and Stryn og Hornindal kommuner (2014). Selected ranking score for each of the scenarios are given in parentheses.

Probability:
- *Frequency of natural hazard:* Once per 10 – 50 years, i.e. probability category D according to Table 2.

Vulnerability assessment:
- *Robustness and buffer capacity:* The waterworks can withstand moderate intensities of debris flows (3)
- *Level of protection:* Partially protected from debris flows (3)
- *Quality level/Age/Level of maintenance and renewal:* Medium age, satisfactory renewal and maintenance (3)
- *Adaptability and quality in operational procedures:* Operated by an experienced operator, ability to adapt to changing framing conditions (2)
- *Transparency/complexity/degree of coupling:* System with large complexity and many interdependencies (4)

Societal consequences:
- *Number of infrastructure users:* 250
- *Duration:* 2-7 days

The abovementioned combination of users and duration qualify for consequence category 4 according to Table 3.
- *Redundancy/substitutes:* Water can be delivered with tank lorry, but at some point after the event the water needs to be boiled to obtain drinking water quality (3)
- *Cascading effects and dependencies*: Moderate cascading effects. (3)
- *Preparedness:* Some risk awareness, simple emergency response plans. (3)
- *Early warning, emergency response and measures:* Limited possibilities for warning (3)





**Scenario 3: Snow avalanche against main road 724 to Oldedalen**

The ranking is based on a similar historic events. Selected ranking score for each of the scenarios are given in parentheses.

Probability

- *Frequency of natural hazard:* More often than once every 10 years, i.e. probability category E according to Table 2.

Vulnerability assessment:

- *Robustness and buffer capacity:* The road will be closed in case of in case of high avalanche danger. (4)
- *Level of protection:* Only partial physical protection against snow avalanches (5)
- *Quality level/Age/Level of maintenance and renewal:* The road has a relative high age, but is satisfactorily maintained (3)
- *Adaptability and quality in operational procedures:* Infrastructure is operated by an experienced operator (2)
- *Transparency/complexity/degree of coupling:* Low degree of coupling. (2)

Societal consequences:

- *Number of infrastructure users:* 100
- *Duration:* 1-2 days

The abovementioned combination of users and duration qualify for consequence category 2 according to Table 3.

- *Redundancy/substitutes:* No alternative roads to Oldedalen (5)
- *Cascading effects and dependencies*: Moderate cascading effects, would affect utility transportation of the Olden mineral water producer (3)
- *Preparedness:* High risk awareness and preparedness regarding snow avalanches (2)
- *Early warning, emergency response and measures:* Limited possibilities and risk reducing effects of warning (3)

**Scenario 4: Storm leading to failure in electricity distribution and communication to the municipal centre**

The ranking is partly based on a similar historic event in December 2011. The selected ranking score for each of the scenarios are given in parentheses.

Probability

- *Frequency of natural hazard:* Severe storms more than once every 10 years. Consideration of  historic frequency of storms and an increase in frequency due to climate change suggests a probability category E in Table 2.

Vulnerability assessment:

- *Robustness and buffer capacity:* Electricity network could withstand storms for some time (3)
- *Level of protection:* Partially protected and well adapted to current climate, but not to future climate (3)

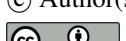


- *Quality level/Age/Level of maintenance and renewal:* increasing age of the components in the electricity network in Norway in general, (Fridheim et al. 2009) (3)
- *Adaptability and quality in operational procedures:* Some ability to adapt to changing framing conditions (3)
- *Transparency/complexity/degree of coupling:* Low degree of coupling (2)

Societal consequences:

- *Number of infrastructure users:* Population in Stryn municipal centre are 2 372, a large number of these could potentially be affected.
- *Duration:* 2-7 days

10 The abovementioned combination of users and duration qualify for consequence category 5 according to Table 3.

- *Redundancy/substitutes:* There exist alternative energy distribution e.g. for critical care facilities, but not for the whole municipality (4)
- *Cascading effects and dependencies*: Considerable importance for societal function (4)
- *Preparedness:* Some risk awareness and preparedness regarding storms. *(3)*
15 
- *Early warning, emergency response and measures:* Storms could be warned, but mitigation actions could potentially only have a small reduction effect on the consequences (3)

**Scenario 5: Landslide against main road E39 at Skredestranda**

The ranking is based on previous historic events, e.g. in November 2015. Selected ranking score for each of the scenarios are given in parentheses.

Probability:

- *Frequency of natural hazard:* More often than once every 10 years, probability category E according to Table 2. (This scenario occurred twice in 2015).

Vulnerability assessment:

- *Robustness and buffer capacity:* The road will be closed in case of a landslide of the considered size. (4)
- *Level of protection:* To a large extent exposed to the event (4)
- *Quality level/Age/Level of maintenance and renewal:* Well-maintained road (2)
- *Adaptability and quality in operational procedures:* Some ability to adapt to changing framing conditions (3)
- *Transparency/complexity/degree of coupling:* Low degree of coupling (2)

Societal consequences:

- *Number of infrastructure users:* >1000





- *Duration:* 2-7 days

The abovementioned combination of users and duration qualify for consequence category 5 according to Table 3.

- *Redundancy/substitutes:* Alternative roads imply major delays (4)
- *Cascading effects and dependencies*: Moderate importance for societal functions (3)
- *Preparedness:* High risk awareness and preparedness regarding snow avalanches (2)
- *Early warning, emergency response and measures:* Limited possibilities and risk reduction effects of warning (3)

**Scenario 6: •   Ice jam breakup in Storelva-river in Hornindal, failure in sewage**

The ranking is partly based on similar historical events. Selected ranking score for each of the scenarios are given in parentheses.

Probability:

- *Frequency of natural hazard:* Every 10 – 50 years, i.e. probability category D according to Table 2.

Vulnerability assessment:

- *Robustness and buffer capacity:* Quite robust, could withstand the event for some time (2)
- *Level of protection:* Partially protected (3)
- *Quality level/Age/Level of maintenance and renewal:* Well-maintained (2)
- *Adaptability and quality in operational procedures:* Experienced operator, ability to adapt to changing framing conditions (2)
- *Transparency/complexity/degree of coupling:* Low degree of coupling (2)

Societal consequences:

- *Number of infrastructure users:* 800
- *Duration:* 2-7 days

The abovementioned combination of users and duration qualify for consequence category 4 according to Table 3.

- *Redundancy/substitutes:* Alternatives which imply disadvantages (3)
- *Cascading effects and dependencies*: Moderate cascading effects (3)
- *Preparedness: Some risk awareness* (3)
- *Early warning, emergency response and measures:* Routines for warning and implementation of measures exist (2)

**Scenario 7: Storm leading to closure of the ferry service between Anda and Lote**

The ranking is based on previous occurrence of this scenario and on information from Stryn og Hornindal kommuner (2014). Selected ranking score for each of the scenarios are given in parentheses.



Probability:

- *Frequency of natural hazard:* More than once every $10^{th}$ year, probability category E according to Table 2.

Vulnerability assessment:

- *Robustness and buffer capacity:* The ferries can operate in strong winds and relatively high waves (3)
- *Level of protection:* To some extent exposed, but well adapted to current climate (3)
- *Quality level/Age/Level of maintenance and renewal:* Well-maintained (2)
- *Adaptability and quality in operational procedures:* Experienced operator, some ability to adapt to changing framing conditions (3)
- *Transparency/complexity/degree of coupling:* Low degree of coupling (2)

Societal consequences:

- *Number of infrastructure users:* 100
- *Duration:* 1-2 days

The abovementioned combination of users and duration qualify for consequence category 2 according to Table 3.

- *Redundancy/substitutes:* Travelers can use alternative roads with small delays (2)
- *Cascading effects and dependencies*: Moderate cascading effects (3)
- *Preparedness:* Emergency response plans are available (3)
- *Early warning, emergency response and measures:* Routines for warning and implementation of measures to limit the consequences exist (2)

**Acknowledgements**

The work described in this paper was supported by the Strategic Project GRAM (GeoRisk Assessment and Management) at NGI. The research leading to these results has also received funding from the European Union Seventh Framework Programme (FP7/2007-2013) under grant agreement n° 606799. The support is gratefully acknowledged.

**Disclaimer**

**The information and views set out in this** paper **are those of the author(s) and do not necessarily reflect the official opinion of the European Union. Neither the European Union institutions and bodies nor any person acting on their**



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

**Tables**





**Table 1 Safety of the population specified through societal values and corresponding consequence types DSB(2014)**

| Safety of the population | |
|---|---|
| **Societal value** | **Consequence type** |
| Life and health | Fatalities |
| | Injuries and diseases |
| Stability | Lack of basic provisions |
| | Disruptions in daily life |
| Nature and environment | Long term damage to the natural environment |
| | Long term damage to the natural environment |
| Material assets | Monetary losses |

**Table 2: Categorization of the probability: Application of the annual probability of the natural event as an initial categorisation of the top event, simplified from the guidelines from the Norwegian Directorate for Civil Protection; DSB (2014). Each category is**
5 **described both with the frequency and the annual probability of the natural event.**

| Category | Frequency of the natural event | Annual probability of the natural event |
|---|---|---|
| E | Higher than once every 10[th] year | > 10% |
| D | Once per 10 - 50 years | 2-10% |
| C | Once per 50 - 100 year | 1-2% |
| B | Once per 100 - 1000 year | 0.1-1% |
| A | Lower than once per 1000 year | < 0,1% |

**Table 3: Initial categorisation of consequence based on the number of infrastructure users and duration of the outage, simplified from the guidelines from the Norwegian Directorate for Civil Protection; DSB (2014). The consequence categories are indicative and**
10 **should be adapted to the municipality's size, i.e. in terms of number of inhabitants.**

| Number of infrastructure users  Duration of the outage/infrastructure loss | < 50 persons | 50-200 persons | 200 – 1000 persons | > 1000 persons |
|---|---|---|---|---|
| > 7 days | 3 | 4 | 5 | 5 |
| 2-7 days | 2 | 3 | 4 | 5 |
| 1-2 days | 1 | 2 | 3 | 4 |





| < 1 days | 1 | 1 | 2 | 3 |
|---|---|---|---|---|

**Table 4 Criteria for ranking of the physical vulnerability indicators and barriers affecting the probability of infrastructure loss. For each indicator, criteria for score values 1-5 are described verbally, where score value 1 corresponds to the lowest vulnerability and 5 to the highest vulnerability.**

| Physical vulnerability indicator | Criteria for choice of score value 1-5 | |
|---|---|---|
| Robustness and buffer capacity | 1 | The infrastructure is robust towards the natural event and/or could withstand the natural event for a long time |
| | 2 | The infrastructure is quite robust towards the natural event and/or could withstand the natural event for some time |
| | 3 | The infrastructure could withstand the natural event if the intensity is low-medium and/or the duration is quite short. |
| | 4 | The infrastructure could only withstand the natural event if the intensity is low and the duration is short. |
| | 5 | The infrastructure is fragile to the natural event. |
| Level of protection (including physical mitigation measures and exposure) | 1 | Infrastructure is not exposed to, or well protected from, the natural event. It is well adapted both to current and future climate. |
| | 2 | Infrastructure has low exposure to or protected from the natural event in study. Well adapted to current climate, partially adapted to future climate. |
| | 3 | Partially protected from the natural event in study. Well adapted to current climate, but not to future climate |
| | 4 | To a large extent exposed to the natural event, insufficiently adapted to current climate |
| | 5 | To a large extent exposed to the natural event, infrastructure is not adapted to current climate |
| Quality level/age/level of maintenance and renewal | 1 | Well-maintained or low age compared with expected lifetime |
| | 2 | Generally well-maintained or relative low age compared with expected lifetime. |





| | 3 | Some planning of renewal and maintenance |
|---|---|---|
| | 4 | Scarce planning of renewal and maintenance. Shortage of resources. |
| | 5 | Corrective maintenance only, ageing infrastructure |
| Adaptability and quality in operational procedures | 1 | Infrastructure is operated by an operator and staff with long experience and/or high ability to adapt to changing framing condition |
| | 2 | Infrastructure is operated by an experienced operator and/or ability to adapt to changing framing condition |
| | 3 | Infrastructure is operated by an operator with some experience and/or some ability to adapt to changing framing condition |
| | 4 | Infrastructure is operated by an operator with very limited experience and/or low ability to adapt to changing framing condition |
| | 5 | The infrastructure is operated by unexperienced operator/staff and/or minimum ability to adapt to changing framing condition |
| Transparency/complexity/degree of coupling | 1 | The system is not dependent on the exposed part of the infrastructure to work and is to a low extent dependent on single components to work. |
| | 2 | The exposed component interacts with a few other components with a low degree of coupling |
| | 3 | The exposed component interacts with many components and the system has a high degree of coupling |
| | 4 | The exposed component is part of a system with a high degree of complexity |
| | 5 | The exposed part of the infrastructure is a component in a system with a high degree of complexity and tight coupling |

**Table 5 Criteria for ranking of societal vulnerability indicators. For each indicator, criteria for score values 1-5 are described verbally, where score value 1 corresponds to the lowest vulnerability and 5 to the highest vulnerability.**




| Societal vulnerability indicator | Criteria for choice of score value 1- 5. | |
|---|---|---|
| Redundancy/substitutes | 1 | There are adequate alternatives or back-up systems for the infrastructure with sufficient capacity |
| | 2 | There are alternatives or back-up systems for the infrastructure which imply few disadvantages for the users |
| | 3 | There are alternatives or back-up systems for the infrastructure, but with limited capacity or which implies disadvantages for the users. |
| | 4 | There exist alternatives, but with low (insufficient) capacity or which imply major disadvantages to the users |
| | 5 | There are no back-up systems nor practical alternatives |
| Cascading effects and dependencies | 1 | The exposed infrastructure has negligible importance for societal functions, no potential cascading effects |
| | 2 | The exposed infrastructure has little importance for societal functions, potentially small cascading effects |
| | 3 | The exposed infrastructure has moderate importance for societal functions, potentially moderate cascading effects. |
| | 4 | The exposed infrastructure has considerable importance for societal functions, potentially considerable cascading effects. |
| | 5 | Important societal functions depend on the exposed infrastructure. Loss of the infrastructure/infrastructure component would potentially have big cascading effects |
| Preparedness | 1 | Very high risk awareness regarding the natural event, exhaustive emergency response plans are available, frequent targeted drills. |
| | 2 | High risk awareness regarding the natural event, emergency response plans are available, targeted drills are performed. |
| | 3 | Some risk awareness regarding the natural event, simple emergency response plans are available |
| | 4 | Weak risk awareness, insufficient emergency response plans |
| | 5 | Lack of risk awareness and knowledge about the natural event, no explicit emergency response plans |
| Early warning, emergency response and measures | 1 | The event is usually predictable well ahead of time and there is enough time for early warning. Thoroughly prepared routines exists for warning |





| | | |
|---|---|---|
| | | and implementation of measures to mitigate the consequences of the natural event. |
| | 2 | The event is usually predictable in time for early warning. There exists routines for warning and implementation of measures to limit the consequences of the natural event. |
| | 3 | The natural event can potentially be predicted, but the routines for warning are insufficient, the warning time is short or mitigation action could potentially only have a small reduction effect on the consequences. |
| | 4 | Low predictability and very short warning time or mitigation action could potentially only have a minor reduction effect on the consequences. |
| | 5 | It is not possible to predict the natural event or there exist no known mitigation measures to limit the consequences. |

**Table 6 Indicative criteria for determining the probability category using vulnerability indicators and adaptation of initial categorization to final categorization**

| Physical vulnerability score | Adjustment of probability category |
|---|---|
| Low (e.g. < 2) | The final probability category is two categories lower than the initial one |
| Medium (e.g. 2-3.5) | The final probability category is one categories lower than the initial one |
| High (e.g. > 3.5) | The final probability category is equal to the initial one |

5  **Table 7 Indicative criteria for determining the consequence category using vulnerability indicators and adaptation of initial categorization to final categorization**

| Societal vulnerability score | Adjustment of consequence category |
|---|---|
| Low (e.g. 1-2.3) | The final consequence category is one category lower than the initial one |
| Medium (e.g. 2.3 – 3.6) | The final consequence category equals the initial one |




| High (e.g. 3.6 – 5) | The final consequence category is one category higher than the initial one |
|---|---|

**Table 8 Ranking of indicators and determination of physical and societal vulnerability scores. The first columns shows the indicator group (i.e. physical or societal vulnerability), the second column, the vulnerability indicators and the next columns, the score values for the scenarios.**

| | | Score values, for scenario no. | | | | | | |
|---|---|---|---|---|---|---|---|---|
| Group | Factor | 1 | 2 | 3 | 4 | 5 | 6 | 7 |
| Vulnerability factors, Vulnerability physical factors, vulnerability of the infrastructure | Robustness and buffer capacity | 4 | 3 | 4 | 3 | 4 | 2 | 3 |
| | Level of protection | 5 | 3 | 5 | 3 | 4 | 3 | 3 |
| | Quality level/Age/Level of maintenance and renewal | 3 | 3 | 3 | 3 | 2 | 2 | 2 |
| | Adaptability and quality in operational procedures | 2 | 2 | 2 | 3 | 3 | 2 | 3 |
| | Transparency/complexity/degree of coupling | 2 | 4 | 2 | 2 | 2 | 2 | 2 |
| | **Average score, physical vulnerability** | 3.5 | 2.9 | 3.5 | 2.9 | 2.9 | 2.2 | 2.7 |
| factors, societal vulnerability | Redundancy/substitutes | 4 | 3 | 5 | 4 | 4 | 3 | 2 |
| | Cascading effects and dependencies | 3 | 3 | 3 | 4 | 3 | 3 | 3 |
| | Preparedness | 1 | 3 | 2 | 3 | 2 | 3 | 3 |
| | Early warning, emergency response and measures | 5 | 3 | 3 | 3 | 3 | 2 | 2 |
| | **Average score, societal vulnerability** | 3.4 | 3.0 | 3.4 | 3.5 | 3.0 | 2.8 | 2.5 |

**Table 9 Initial and final categorisation of probability and consequence. The difference between the final and initial probability category is determined by the physical vulnerability score. The difference between the final and initial consequence category is determined by the societal vulnerability score.**

| Group | | Sc. 1 | Sc. 2 | Sc. 3 | Sc. 4 | Sc. 5 | Sc. 6 | Sc. 7 |
|---|---|---|---|---|---|---|---|---|
| Probability | Initial probability category according to Table 2 | E | D | E | E | E | D | E |





| | | E | C | E | D | E[(i)] | C | D |
|---|---|---|---|---|---|---|---|---|
| | Final probability category according to physical vulnerability scores in Table 8 and criteria in Table 6. | E | C | E | D | E[(i)] | C | D |
| Consequence | Number of infrastructure users | 800 | 250 | 100 | >1000 | >1000 | 800 | 100 |
| | Duration of the outage/infrastructure loss (days) | 1-2 | 2-7 | 1-2 | 2-7 | 2-7 | 2-7 | 1-2 |
| | Initial consequence category according to Table 3 | 3 | 4 | 2 | 5 | 5 | 4 | 2 |
| | Final consequence category according to societal vulnerability scores in Table 8 and criteria in Table 7. | 3 | 4 | 3 | 5 | 5 | 3 | 2 |

(i)     This probability category was not adjusted downwards even if the physical vulnerability score would indicate that. The reason is that the actual landslide probability is much higher than the lower limit of the probability category.



**Table 10 Results from the semi-quantitative analyses**

| | 1 | 2 | 3 | 4 | 5 |
|---|---|---|---|---|---|
| E | | | Sc. 1 & 3 | | Sc. 5 |
| D | | Sc. 7 | | | Sc. 4 |
| C | | | Sc. 6 | Sc. 2 | |
| B | | | | | |
| A | | | | | |

Probability / Consequence

**Risk level**

| | |
|---|---|
| *7* | |
| *6* | |
| *5* | |
| *4* | |
| *3* | |
| *2* | |
| *1* | |