# Peer review of "Assessing the risk posed by natural hazards to infrastructures"

_Natural Hazards and Earth System Sciences, 2016_

## Referee Comment (RC1) · M. Papathoma-Koehle (Referee) · 2 May 2016

The paper proposes a method for risk assessment of infrastructures to a wide range of natural hazards. The article certainly addresses a topic which is well within the scope of NHESS, however, considerable changes should be made prior to publication which will better highlight the value of the work which is presented here.

Comments:

1. Introduction and Background chapter: The authors refer to three levels of analysis. How do these levels correspond to the aims/end users of the method presented here?

2. Figure 1: What is a "Top event"? What are the "Barriers"? What do the authors mean with "loss of infrastructure"? is this 100% loss? Or just disruption?

[Figure]

3. The theory and literature review part that should be found in the Introduction/Background chapter is rather weak. In more detail I have the following comments:

a. The authors refer to "critical infrastructure" (p.2/l.11). What is critical infrastructure and what does it include? The authors should consider providing some basic definitions. In this way terms that are incorrect or need to be defined and can be found throughout the text may be avoided e.g. "the severity of risk" (p.2/l.18), "what can go wrong (evaluation of sensitivity (susceptibility) and resilience)" (p.2/l.25), "acceptability/tolerability of risk" (p.2, l.30), "susceptibility functions" (p.4, l.23), "mapping threats of natural hazards" (p.3,l.11).

b. A very important topic in the risk analysis of infrastructure is "resilience". The authors refer to this term only once (p.2, line 25) without explicitly discussion it. Studies regarding the resilience of infrastructure are also not mentioned.

c. The literature review can be found in page 4, lines 17-30, however, the authors list the papers that contain reviews on the topic "vulnerability and infrastructure" without giving any additional information. What is the state of the art of physical and/or social vulnerability assessment regarding infrastructure? What kind of methods have been proposed and what is used by authorities and decision makers?

4. Figure 2 is the core figure of the paper presenting the methodology. In my opinion, the Figure should be revised in order to better show:

a. The three steps described in the methodology chapter (4.1).

b. The grouping of the indicators (physical/social vulnerability). The authors claim that the physical vulnerability indicators are demonstrated in Figure 2 (p.9/l.19-20)

5. The methodology has to be described in greater detail focusing on the following topics:

a. In Table 4 the criteria for choice of score are described in a very trivial way. "The infrastructure is robust towards the natural event and/or could withstand the natural

event for a long time/some time/quite short or short duration". How do the authors define or differentiate between, for example, "short" and "quite short" duration? Is this the same for all scenarios? (what is short duration for a road closure is not short for electricity distribution etc.)

b. What about the weighting of the indicators? In the discussion chapter you refer to a "flexible weighting system" (p.15, l.30-31). Where is this system described? Which weighting method do you use? Are all indicators equally important?

c. Risk ranking: The risk ranking seems quite arbitrary to me. Who and based on which criteria identifies the risk levels? Often the risk levels are identified by decision makers and are connected to specific actions (e.g. evacuation). Is this the case here?

6. Application chapter: The application chapter may be significantly improved if the authors consider the following:

a. Figure 3: The authors should consider to improve the figure by adding a small map of Norway showing the location of the case study within the country. Moreover, they should consider starting the session with text and not the figure itself.

b. The session may be illustrated with more information regarding the case study area. Why was it chosen? Have they experienced the impact of natural hazards in the past affecting their infrastructure? How many people live there? Is the infrastructure important for the local community or for the whole country?

c. Listing the generic scenarios and then the site-specific scenarios seems like a repetition to me. I would just keep the second list with the numbered site specific scenarios.

7. Results: The results are summarized in Table 8. The seven scenarios are assigned with a risk level. Is the aim of the study to compare these scenarios and if yes is this really possible? (see previous comment on criteria for choice of scores). If the aim of the paper is among others the comparison, be consistent and refer to it in the scope chapter.

8. The discussion chapter is too short. I would expand it making two sessions: one session highlighting the usefulness and advantages of the methods through examples and one session outlining the assumptions that had to be made, the limitations but also the future developments that are necessary. The authors refer to many interesting topics that need to be further discussed and sometimes illustrated with examples:

a. P.14/l. 28-29: "It provides more guidance to the user than general risk assessment methods": Which other methods are available and what does this method offer that the others cannot (link to literature review)? This needs to be further discussed here in order to highlight the benefits of your method.

b. P.15, l.1: Give an example showing the usefulness of this ranking. As mentioned before the ranking may be connected to specific actions/decisions.

c. P. 15, l. 11: "implicit guidance on how to reduce vulnerability". This is an interesting topic and a great opportunity for the authors to highlight the usefulness of the method. Please give some examples on how can this be achieved by using the specific approach.

9. Appendix: In the main text the authors refer to the assessment of two dimensions of vulnerability: Physical and societal. However, in the appendix they refer to "vulnerability assessment" and "societal consequences". Is this the same? (See comment about definitions).

10. General comment: be careful with the use of the word "infrastructure" in plural. I am not sure if it really exists (I am not an English native speaker myself). Please check.

Please also note the supplement to this comment:
http://www.nat-hazards-earth-syst-sci-discuss.net/nhess-2016-89/nhess-2016-89-RC1-supplement.pdf

---

## Referee Comment (RC2) · Anonymous Referee #2 · 26 Jul 2016

The paper deals with risk related to interaction between natural phenomena and critical infrastructures which is a key issue in risk management process. The authors propose a two level empirical methodology to assess this risk. Despite of its operational interest, the major issue is that the whole methodology is fully empirical with very few references to existing works in such domains. Therefore, despite of some good ideas, it is quite difficult to trust in the method and its results : many subjective choices are done without being clearly explained and described. Using and applying the method would be difficult and it is now clear to see how this process can be generalized. What is the added value in comparison with decision making methods, safety analysis already used to assed criticality of interdependent infrastructures ? A detailed review including remarks is proposed in the annotated pdf. Some example of issues in the text relate are described below : 1. Insufficient definition of concepts used (risk,

hazard, phenomena, uncertainty on risk, potential risk, societal vulnerability ? ...). Some definitions are contradicting with state of the art : this has to be changed or discussed completely. Why are new definitions proposed ? what do they correspond to ? How are they justified ? 2. Studied infrastructures are not described 3. Figures are not informative (eg figure 1 , what about existing cause effect consequence diagram) 4. Not enough references to existing frameworks related 1) to safety and reliability analysis (functional analysis, failure modes etc...) 2) to classical decision-aiding methods such as multicriteria decision making methods : proposed aggregation is a weighted average, why are there no references to classical aggregation methods (MCDM)? 5. Some keys issues about choosing criteria are not described 6. Some tables are not understandable (eg table 5) 7. The calculation process robustness itself is not tested and described. The adjustment are not understandable. How can vulnerability be used to modify the frequency of a phenomenon ?!!! "The physical vulnerability score is used to adjust the probability category assessed in the initial categorization and the societalvulnerability score is used to adjust the consequence category assessed in the initial categorization"

Please also note the supplement to this comment: http://www.nat-hazards-earth-syst-sci-discuss.net/nhess-2016-89/nhess-2016-89-RC2-supplement.pdf

**Supplement:**

[revised manuscript text omitted]

---

## Author Comment (AC1) · 6 Sep 2016

Reply to RC1: Thanks a lot for your valuable comments and your specific and fruitful suggestions for improvements for the paper. I appreciate the effort you have put into this! Replies to your overall comments and questions follow in the supplement file.

*The paper proposes a method for risk assessment of infrastructures to a wide range of natural hazards. The article certainly addresses a topic which is well within the scope of NHESS, however, considerable changes should be made prior to publication which will better highlight the value of the work which is presented here.*

*Comments:*

*1. Introduction and Background chapter: The authors refer to three levels of analysis. How do these levels correspond to the aims/end users of the method presented here?*

More information about the overall municipal vulnerability and risk analysis, which this method is intended to be applied within will be provided and it will be demonstrated which part of the process the method is intended to cover.

For your information:

The overall municipal vulnerability and risk analysis, which this method is intended to cover parts of is performed as a project with a project team, project leader and steering committee anchored in the municipality's administration and political leadership. The analysis consists of the following stages:

1. Identification of adverse events,(considering threats within the municipality within the municipality or outside the municipality, but with consequences for the municipality)

2. Assessing risk and vulnerability of adverse events,

3. Providing an overview of the risks associated with each of the identified adverse events (from point 1)

4. Following-up and

5. Reporting.

Level 1 of the model in this article is aimed at the first stage: Identification of adverse events, levels 2 and 3 of the model target the second and third stages in the municipal risk and vulnerability analyses: assessing risk and vulnerability of adverse events and providing an overview of the risks associated with each of the identified event in the municipality. The second level of the model(i.e. the explicit model proposed in this paper) is used to give a coarse overview of the risks used for preliminary sorting of the events, while the third level is used for a more detailed analysis of the events as background for decisions regarding follow-up. For the level 2 analysis, it is chosen not to include explicit criteria or risk thresholds for recommendations regarding the follow-up, both because each municipality must adapt the criteria for follow-up to their own situation and capacity (scenarios with the highest risk must be prioritised regardless of risk acceptance) and because the method is a rough analysis where the scale is relative and is too coarse to make decisions regarding risk acceptance.

*2. Figure 1: What is a "Top event"? What are the "Barriers"? What do the authors mean with "loss of infrastructure"? is this 100% loss? Or just disruption?*

Figure 1 shows a bow-tie diagram which shows both the connection between and difference between proactive and reactive risk management, showing causes of the infrastructure malfunctioning on the left side and consequences of this malfunctioning on the right side. The analysed scenario/adverse event is given in the middle. It is often referred to as the Top Event

The Top Event is when actual damage is caused by the hazard, i.e. malfunctioning of the infrastructure in this paper.

The scenarios could be controlled using Barriers both for reduction of the probability of the Top Event/prevention controls, i.e. barrier which could prevent causes of the Top event and barriers for mitigation and recovery controls, i.e. barriers that limit the consequences of the Top Event.

Barriers, could be physical, or organisational including human behaviour. Barriers may prevent the Top Event (Malfunctioning of infrastructure) and mitigate its consequences.

In the revised version of the paper the terms "Adverse Event will be used instead of "Top Event" (to avoid unnecessary introduction of new terms) and , Malfunctioning of infrastructure will be used instead of "Loss of infrastructure" to make it clearer that it refers to an interruption(partly or fully) of the services provided by the infrastructure.

Section 2 will be extended to include both scope and terminology and the terminology modified accordingly to be consistent within the paper.

*3. The theory and literature review part that should be found in the Introduction/Background chapter is rather weak. In more detail I have the following comments:*

*a. The authors refer to "critical infrastructure" (p.2/l.11). What is critical infrastructure and what does it include? The authors should consider providing some basic definitions. In this way terms that are incorrect or need to be defined and can be found throughout the text may be avoided e.g. "the severity of risk" (p.2/l.18), "what can go wrong (evaluation of sensitivity (susceptibility) and resilience)" (p.2/l.25), "acceptability/tolerability of risk" (p.2, l.30), "susceptibility functions" (p.4, l.23), "mapping threats of natural hazards" (p.3,l.11).*

Terminology will be included in the extended section 2: Scope and terminology. Terminology will be reviewed and simplified if possible to avoid unnecessary introduction of new terms

*b. A very important topic in the risk analysis of infrastructure is "resilience". The authors refer to this term only once (p.2, line 25) without explicitly discussion it. Studies regarding the resilience of infrastructure are also not mentioned.*

Vulnerability and resilience are closely related and some authors, including Adger (2000), view resilience and vulnerability as equivalent but opposite concepts. The paper has focus on vulnerability, but also on how to reduce the vulnerability, which belongs more naturally to the resilience part. The text will be modified to refer to both vulnerability and resilience. In addition, studies that specifically address resilience will be included.

*c. The literature review can be found in page 4, lines 17-30, however, the authors list the papers that contain reviews on the topic "vulnerability and infrastructure" without giving any additional information. What is the state of the art of physical and/or social vulnerability assessment regarding infrastructure? What kind of methods have been proposed and what is used by authorities and decision makers?*

A summary of the literature which is referred to will be included answering these questions. In addition, some key references on resilience will be reviewed.

*4. Figure 2 is the core figure of the paper presenting the methodology. In my opinion, the Figure should be revised in order to better show:*

*a. The three steps described in the methodology chapter (4.1).*

*b. The grouping of the indicators (physical/social vulnerability). The authors claim that the physical vulnerability indicators are demonstrated in Figure 2 (p.9/l.19-20)*

Thank you for this suggestion! The figure has been improved to better show points a – b; the new version is shown in Figure below. Also the terminology has been modified to be consistent through the paper. The meaning of "Risk posed by natural hazards to infrastructure" need to be explained as well, i.e. that the focus are on the indirect losses and consequences in terms of "loss of stability"

*5. The methodology has to be described in greater detail focusing on the following topics:*

*a. In Table 4 the criteria for choice of score are described in a very trivial way. "The infrastructure is robust towards the natural event and/or could withstand the natural event for a long time/some time/quite short or short duration". How do the authors define or differentiate between, for example, "short" and "quite short" duration? Is this the same for all scenarios? (what is short duration for a road closure is not short for electricity distribution etc.)*

Improved description of buffer capacity which is linked to the duration of the natural event will be provided, based on typical durations of the natural hazard, for the intensity and frequency analysed in the current scenario. The meaning of short and long duration depends, in addition to the type of the natural hazard, on the type of infrastructure. However, the focus for this indicator is on its effect of the probability of an infrastructure malfunctioning, thus the focus is in terms of duration related to the hazard type only. (The duration of the malfunctioning is included in the assessment of the severity of consequences for the society)

Suggestion:

1: Highest buffer capacity: Could withstand the natural event for a duration more than 2 times the median duration of the natural event

2: High buffer capacity: 1 - 2 times the median duration of the event

3: Medium buffer capacity: 0.5 – 1 times the median duration

4: Low buffer capacity: less than 0.5 times the median duration

5: Lowest: Fragile.

Improved also description of age: low age = less than 10%, relative low age 10-20% of expected lifetime (based on a note in Norwegian about lifetime and maintenance needs for roads and railways: Simonsen, M. (2010): Levetid og lengde for vei og jernbane, Notat 10 januar 2010, Vestlandsforsking)

1: Well-maintained or age is < 15% of expected lifetime

2: Generally maintained or age is 15-30% of expected lifetime

*3-5: unchanged.*

*b. What about the weighting of the indicators? In the discussion chapter you refer to a "flexible weighting system" (p.15, l.30-31). Where is this system described? Which weighting method do you use? Are all indicators equally important?*

The weighting system is only shortly mentioned as a linear weighted average. A more thorough explanation will be included.

Some explanation to the weighting system – for your information:

A weighting system is introduced to account for the relative importance of each indicator for the total vulnerability level. If all the indicators are believed to be of equal significance, equal weighting should be applied. Techniques to determine weights include expert judgment, the analytical hierarchy process, principal component analysis and factor analysis (CIMNE 2009). In this work, the weights are to be chosen by the user of the method based on experience and local knowledge.

Each indicator is weighted based on its overall degree of influence. The weights are to be chosen among the values 1(least influential), 2(moderately influential) and 3(most influential). When all the indicators are assigned a vulnerability score, the score for each indicator is multiplied with its corresponding weight and summed to give a weighted vulnerability score. The final vulnerability estimate is formulated as a weighted average of the individual indicator scores

Vulnerability index= $\sum_{\text{All indicators}}$ indicator score $\cdot$ indicator weight$/\sum_{\text{All indicators}}$ indicator weight

A possible improvement of the method would be to use Analytical Hierarchy Process (AHP) in the weighting of the vulnerability indicators.

*c. Risk ranking: The risk ranking seems quite arbitrary to me. Who and based on which criteria identifies the risk levels? Often the risk levels are identified by decision makers and are connected to specific actions (e.g. evacuation). Is this the case here?*

Even if the vulnerability is assessed relatively, the initial classification is quantitative and each cell could therefore be anchored in quantitative risk estimates. By applying the quantitative criteria as basis to assign a risk range to each cell in the risk matrix, it may be shown that the diagonal lines in the risk matrix approximate represent equivalent risk levels, i.e. on a gross scale, the risk is equal along diagonal lines. The approach is useful for prioritization of mitigation measures e.g. give priority to a certain sector instead of another one because it can lead to more severe effects.

It is chosen not to include explicit criteria or risk thresholds for recommendations regarding the follow-up and risk acceptance, both because each municipality must adapt the criteria for follow-up to their own situation and capacity (scenarios with the highest risk must be prioritised regardless of risk acceptance) and because the method is a rough analysis where the scale is relative and is too coarse to make decisions regarding risk acceptance.

Uncertainty in classification and subdivision will be discussed in the discussion chapter.

*6. Application chapter: The application chapter may be significantly improved if the authors consider the following:*

*a. Figure 3: The authors should consider to improve the figure by adding a small map of Norway showing the location of the case study within the country. Moreover, they should consider starting the session with text and not the figure itself.*

Agree!

*b. The session may be illustrated with more information regarding the case study area. Why was it chosen? Have they experienced the impact of natural hazards in the past affecting their infrastructure? How many people live there? Is the infrastructure important for the local community or for the whole country?*

The reasoning for the choice of case study area will be included. The study area is exposed to different types of natural hazards, especially landslides, but also floods and storms, which need to be considered during the development of infrastructure and residential and commercial buildings in the municipality. The second author is knowledgeable about the hazard situation in the area and has been involved with the municipal analyses in this area. Natural hazards have affected infrastructure repeatedly in the past and data from previous events (e.g. frequency of natural events and duration of infrastructure outage) could be used to test the method, as shown in the "Application examples for the municipalities Stryn and Hornindal" – session.

*c. Listing the generic scenarios and then the site-specific scenarios seems like a repetition to me. I would just keep the second list with the numbered site specific scenarios.*

I agree, the first list will be deleted.

*7. Results: The results are summarized in Table 8. The seven scenarios are assigned with a risk level. Is the aim of the study to compare these scenarios and if yes is this really possible? (see previous comment on criteria for choice of scores). If the aim of the paper is among others the comparison, be consistent and refer to it in the scope chapter.*

Yes, the purpose of this explicit method is comparison between scenarios and identification of the most critical amongst them. The method is referred to as a screening tool in the scope, but the text will be expanded to better explain that the comparison of scenarios is an important part of "screening tool". It is definitely a difficult task, which usually would involve extensive use of subjective judgment, but the anchoring in the quantitative estimates justifies the comparison. Also, by law, the Norwegian municipalities are required to carry out risk and vulnerability analyses across sectors in the municipality, providing an overview of the risks associated with each of the identified adverse events and to set priorities based on the results.

*8. The discussion chapter is too short. I would expand it making two sessions: one session highlighting the usefulness and advantages of the methods through examples and one session outlining the assumptions that had to be made, the limitations but also the future developments that are necessary.*

Agree. The discussion chapter will be extended and subdivided into two subsessions according to your suggestion. The usefulness and added value will be described according to the points a-c below.

For the limitations and future work-session uncertainty in classification and final ranking will be discussed, as well as the need for calibration and the challenges in calibration of the type of risk which is linked to the indirect losses, which cannot be measured directly, but has to be estimated.

*The authors refer to many interesting topics that need to be further discussed and sometimes illustrated with examples:*

*a. P.14/l. 28-29: "It provides more guidance to the user than general risk assessment methods": Which other methods are available and what does this method offer that the others cannot (link to literature review)? This needs to be further discussed here in order to highlight the benefits of your method.*

Examples of the explicit guidance will be included in the text. Main advantage of the methodology is that it could be used by non-experts, in particular the stakeholders in the municipality. It is desirable that the municipality leads this analysis themselves, but with relevant experts in the team.  It guides the user in which vulnerability factors to assess, both for assessment of the probability of the infrastructure malfunctioning and the societal consequences, it has explicit criteria for how the indicator contributes to the overall vulnerability through the explicit ranking criteria and how aggregate the results, even if some judgment is required when using the method.

*b. P.15, l.1: Give an example showing the usefulness of this ranking. As mentioned before the ranking may be connected to specific actions/decisions.*

The purpose of the municipal vulnerability and risk analysis is among others to provide an overview of adverse events that challenges the municipality, assess risk and vulnerability across sectors and provide a basis for goals, priorities and the necessary decisions in the municipality work with Civil Protection and Emergency. It is also within the responsibility of the municipalities to help maintain critical societal functions also when adverse events occur. The screening tool proposed in this paper assigns a risk levels to each of the analysed scenario/adverse events. It provides a useful basis to prioritise between the following-up of the different scenarios, where accordingly the scenarios with the highest risk levels should be analysed further and followed up.  See also point 5c.

*c. P. 15, l. 11: "implicit guidance on how to reduce vulnerability". This is an interesting topic and a great opportunity for the authors to highlight the usefulness of the method. Please give some examples on how can this be achieved by using the specific approach.*

The method assesses several aspects of vulnerability and resilience and could, in addition to the risk ranking be used to identify the indicators contributing most to the vulnerability for each case. Special focus should be paid to indicators with a poor score and of high importance, i.e. a high weight.  The identification of the most critical indicators could be used as a guideline where to focus the further effort. Example from the case studies on this will be included.

*9. Appendix: In the main text the authors refer to the assessment of two dimensions of vulnerability: Physical and societal. However, in the appendix they refer to "vulnerability assessment" and "societal consequences". Is this the same? (See comment about definitions).*

The two dimensions of vulnerability will be modified to physical and socio-economic. In the appendix, the assessment was to be subdivided into two parts: 1) assessment of the probability of malfunctioning of the infrastructure (consisting of assessment of the probability of the natural event as well as assessment of the physical vulnerability indicators) and 2) assessment of the societal consequences (consisting of assessment of number of infrastructure users and duration as well as assessment of the socio-economic vulnerability indicators). The titles describing the assessment

steps will be modified accordingly to make this clearer and to be consistent with the terminology in the rest of the paper.

*10. General comment: be careful with the use of the word "infrastructure" in plural. I*

*am not sure if it really exists (I am not an English native speaker myself). Please check.*

In Oxford Learner's Dictionary, could be classified as "countable, uncountable." http://www.oxfordlearnersdictionaries.com/definition/english/infrastructure

It is in literature often referred to in plural, but it could be that these are written by non-native writers. I will consider to use infrastructure-sectors instead when the need to use the word "infrastructure" in plural.

---

## Author Comment (AC2) · 6 Sep 2016

Thank you for your valuable comments and questions to the paper in the review, as well as the detailed comments given within the paper itself. We tried to make the paper as short as possible, but after the review, I realize that the steps in the methodology were a bit scarcely described and justified and that some parts need to be extended to better explain and reason the choices made. In addition, the paper applies and refers to different references with slightly different use of terminology. Terminology used in the paper will therefore be reviewed and uniform use ensured. Replies to your overall comments and questions follow in the supplement - or below:

Thank you for your valuable comments and questions to the paper in the review, as well as the detailed comments given within the paper itself. We tried to make the paper

as short as possible, but after the review, I realize that the steps in the methodology were a bit scarcely described and justified and that some parts need to be extended to better explain and reason the choices made. In addition, the paper applies and refers to different references with slightly different use of terminology. Terminology used in the paper will therefore be reviewed and uniform use ensured. Replies to your overall comments and questions follow below:

1a) "The paper deals with risk related to interaction between natural phenomena and critical infrastructures which is a key issue in risk management process. The authors propose a two level empirical methodology to assess this risk. Despite of its operational interest, the major issue is that the whole methodology is fully empirical with very few references to existing works in such domains. Therefore, despite of some good ideas, it is quite difficult to trust in the method and its results : many subjective choices are done without being clearly explained and described. Using and applying the method would be difficult and it is now clear to see how this process can be generalized."

Reply: Semi-quantitative, indicator-based methods will necessarily require use of (expert) judgment and accordingly be associated with subjectivity and uncertainties. Indicators are commonly used in vulnerability and resilience assessment, since it is often difficult to quantify vulnerability and resilience in absolute terms without any external reference with which to validate the calculations. Indicators are typically used to assess relative levels of vulnerability and resilience either to compare between places, or to analyse trends over time. In this paper, an indicator-based approach is combined with an initial quantitative categorization, based on explicit quantitative criteria to limit the uncertainty and effect of subjective judgment on the results. The limitations of the method, sources of uncertainty and needs for calibration will be discussed in the discussing session. Uncertainties are related to properties of indicator-based methods in general (as mentioned above); and to the scope of the method, which are applicable for different infrastructure sectors and uses generic factors for infrastructure vulnerability.

Some explanations to the indicator –based part of the methodology, for your information: The chosen indices reflect different aspects of vulnerability and resilience of infrastructures. The choice of generic indicators relevant for the probability of infrastructure malfunctioning and for the societal consequences of the malfunctioning infrastructure is in accordance with what is documented in literature as reviewed in the Background-section (Section 3) . The ranking of the indicators are based on their relevance for the probability of a malfunctioning of the infrastructure or for the societal consequences of the malfunctioning. The optimal realization of an indicator (i.e. it's lowest possible contribution to vulnerability) and the least favorable (most unfavorable) realization of an indicator (i.e. its highest possible contribution to the vulnerability) follows implicitly from the reasoning of the choice of that specific indicator. For instance, for the "Redundancy/substitutes" – indicator, the optimal realization is if there exists (or is possible to establish) an alternative that provide the same service as the analysed piece of infrastructure. The most unfavorable realisation is if there are no other way to provide the same service as the analysed piece of infrastructure. The optimal realisation of an indicator is given score 1, while the most unfavorable realization is allocated a score 5. Based on the description of these extremes, one could then leave to the user to decide a score between 1-5 where 1 corresponds to the optimal realisation and 5 to the most unfavorable realisation. However, to limit the use of subjective judgment of the user and to make the method easy to use, 3 levels between the two extremes were also defined, with corresponding descriptions of what the realisation of the indicator within each level implies. The user need to choose between integer values 1-5 according to the description.

A subjective choice left to the user is the choice of weights for each indicator. The weights should e.g. be chosen in accordance with the type of infrastructure, site-specific factors and conditions etc. (The indicators are aggregated through a weighted linear average).

1b) "What is the added value in comparison with decision making methods, safety analysis already used to assed criticality of interdependent infrastructures ? "

Reply: There is no all-encompassing method available to analyse all aspects of critical infrastructures, but different methods serve different purposes and have different advantages (and disadvantages). The advantages with use of the proposed method is that it is generic and has a very broad scope (applicable for assessment of socioeconomic risk associated with malfunctioning in different infrastructure sectors), it could be used by local stakeholders as a supporting tool when performing the municipal risk and vulnerability analysis, it is very easy to use, and gives results that are easy to communicate (a risk level). Indicators are useful for reducing complexity, measuring progress, mapping and setting priorities and they could serve as an important tool for decision makers. It serves well the purpose of being a screening tool for scenarios of natural events threatening critical infrastructure in a municipal risk and vulnerability analysis, even if is not feasible for more detailed studies of the risk. Advantages of the method will be discussed in the discussion section, extended according to suggestions from RC1.

1c) "Some example of issues in the text relate are described below: 1. Insufficient definition of concepts used (risk, hazard, phenomena, uncertainty on risk, potential risk, societal vulnerability ? : : :). Some definitions are contradicting with state of the art : this has to be changed or discussed completely. Why are new definitions proposed ? what do they correspond to? How are they justified ?"

Reply: I agree that issues related to the terminology need to be improved. A separate chapter discussing terminology and scope will be included. (i.e by extending the existing chapter 2.) It should be clearer that the focus of this method is on the indirect losses, focusing on loss of stability for the exposed population. It should also contain the different dimensions of vulnerability referred to in the paper, i.e., physical, social and economical.

I assume, based on comments within the paper, that the "definitions contradicting with the state of the art" – issues are referring to Figure 2 and the decomposing of risk into the probability/frequency of infrastructure malfunctioning caused by natural hazard and

the societal consequences following from the malfunctioning infrastructure. The figure shows that risk could be decomposed into probability and consequence as in traditional risk definitions, but a clarification of what the probability and consequences encompass is needed. The methodology presented in this paper is adapted to be in accordance with the guidelines of the Norwegian Directorate for Civil protection, DSB(2014). In these guidelines, the addressed probability is the probability of an adverse event involving material destruction, i.e. not the probability of the natural event. The adverse event could e.g. involve a natural event leading to a damage on the infrastructure (in this paper: destruction leading to malfunctioning of infrastructure). The considered consequences are the societal consequences of the adverse event, here: the malfunctioning infrastructure. Similar subdivisions are found in DSB(2014), Lenz(2009), IRGC(2007), and http://www.nap.edu/read/6425/chapter/4#16: "It is useful to distinguish between the physical destruction caused by natural disasters to human beings and property, and the consequences of that destruction." This also corresponds to subdivisions of the strategies for risk reduction into strategies that minimise the probability of infrastructure failure, and those that minimise the negative effects of a failure, (IRGC, 2007). It will be made clearer that the considered risk assessment focuses on the indirect losses and the consequences referred to are in terms of the societal value "Stability".

2. "Studied infrastructures are not described "

Reply: The infrastructure sectors studied/applicable for in the method are Electric power supply; Transportation; Urban water supply and wastewater treatment; ICT systems. However, the focus will be on the three first, as EWE and natural hazards are less relevant direct cause of malfunctioning for ICT systems. These infrastructures were chosen as they provide the essential functions and services of a society. They also share a number of similarities. Information about this will be included into the paper.

3. "Figures are not informative (eg figure 1 , what about existing cause effect consequence diagram)"

Reply: Figure 1 has the same structure as simple cause and effect diagram – and the cause and effect diagram used by DSB(2014). The main purpose of the figure is to define the scope and terminology. The terminology will be reviewed and if possible simplified and unified. Also figure 2 will be improved according to suggestions form RC 1.

4. "Not enough references to existing frameworks related 1) to safety and reliability analysis (functional analysis, failure modes etc: : :) 2) to classical decision-aiding methods such as multicriteria decision making methods : proposed aggregation is a weighted average, why are there no references to classical aggregation methods (MCDM)? "

Reply: Paper touches upon several broad topics such as, vulnerability assessment of critical infrastructure, risk assessment, assessment of direct and indirect losses. It might be too detailed for the scope of the method to describe safety and reliability analysis as well as decision theory, but I will look into framework that could be shortly referred to and discussed briefly.

5. "Some keys issues about choosing criteria are not described "

Reply: More details about the ranking criteria will be added. See comment about ranking criteria in point 1a) as well as description in reply to RC1, point 5a).

6. "Some tables are not understandable (eg table 5) "

Reply: I am not sure which part Table 5 is not understandable, but assume that the comment is referring to the ranking criteria? See explanations given to point 1a).

7. "The calculation process robustness itself is not tested and described. The adjustment are not understandable. How can vulnerability be used to modify the frequency of a phenomenon ?!!! "The physical vulnerability score is used to adjust the probability category assessed in the initial categorization and the societalvulnerability score is used to adjust the consequence category assessed in the initial categorization""

Reply: Calculation process description and reasoning will be included.

Adjustment of the probability of infrastructure malfunctioning:

The adjustment represent the step from the P(natural event), used in the initial categorization to P(infrastructure malfunctioning caused by natural event), which is assessed in the final categorisation. The idea behind this adjustment was to use the physical vulnerability index as a proxy for the conditional probability of infrastructure malfunctioning given/under condition that the natural event has occurred.

P(infrastructure malfunctioning caused by natural event) = P(natural event) * P(infrastructure malfunctioning|natural event)

If the infrastructure has a high physical vulnerability index, then the conditional probability P(infrastructure malfunctioning|natural event) is high, i.e. approximately 1; and thus we get the equation:

P(infrastructure malfunctioning caused by natural event) $\approx$ P(natural event)

Then the intitial hazard class is kept. (The frequency of the natural event is used for initial categorization of the frequency of the infrastructure malfunctioning) On the other extreme, if the physical vulnerability index is very low; the conditional probability, P(infrastructure malfunctioning|natural event) is low, e.g. in the order of 0.1, then the relation yields:

P(infrastructure malfunctioning caused by natural event) $\approx$ 0.1 * P(natural event)

Accordingly, a multiplication of the probability with 0.1 corresponds to a reduction in probability category with 1-2 probability categories, i.e. P(infrastructure malfunctioning caused by natural event) is 1-2 probability categories lower than P(natural event). The step from the P(natural event), used in the initial categorization and P(infrastructure malfunctioning caused by natural event) assessed in the final categorization is the background for adjustment of the probability categories.

Adjustment of the consequence classes:

Similarly, the number of people affected by the malfunctioning infrastructure could be higher or lower than the number of infrastructure users, dependent on how the situation is handled and how important the malfunctioning infrastructure is for the society. The socio-economic vulnerability index is a proxy for the societal capacity to cope with malfunctioning infrastructure. Accordingly, if the socio-economic vulnerability index is low, then the number of affected people will be lower than the number of infrastructure users, e.g. if the infrastructure malfunctioning is managed well and substitutes for the service provided by the malfunctioning infrastructure established. However, if the socio-economic vulnerability index is high, then the number of affected will be higher than the number of infrastructure users, e.g. if there are large cascading effects.

Please also note the supplement to this comment:
http://www.nat-hazards-earth-syst-sci-discuss.net/nhess-2016-89/nhess-2016-89-AC2-supplement.pdf